# Double thermoelectric power factor of a 2D electron system

Yuqiao Zhang[1], Bin Feng[2], Hiroyuki Hayashi[3], Cheng-Ping Chang[4], Yu-Miin Sheu[4,5], Isao Tanaka[3], Yuichi Ikuhara[2] & Hiromichi Ohta [1,6]

Two-dimensional electron systems have attracted attention as thermoelectric materials, which can directly convert waste heat into electricity. It has been theoretically predicted that thermoelectric power factor can be largely enhanced when the two-dimensional electron layer is far narrower than the de Broglie wavelength. Although many studies have been made, the effectiveness has not been experimentally clarified thus far. Here we experimentally clarify that an enhanced two-dimensionality is efficient to enhance thermoelectric power factor. We fabricated superlattices of [$N$ unit cell SrTi$_{1-x}$Nb$_x$O$_3$|11 unit cell SrTiO$_3$]$_{10}$—there are two different de Broglie wavelength in the SrTi$_{1-x}$Nb$_x$O$_3$ system. The maximum power factor of the superlattice composed of the longer de Broglie wavelength SrTi$_{1-x}$Nb$_x$O$_3$ exceeded $\sim$5 mW m$^{-1}$ K$^{-2}$, which doubles the value of optimized bulk SrTi$_{1-x}$Nb$_x$O$_3$. The present approach—use of longer de Broglie wavelength—is epoch-making and is fruitful to design good thermoelectric materials showing high power factor.

[1] Graduate School of Information Science and Technology, Hokkaido University, N14W9, Kita, Sapporo 060−0814, Japan. [2] Institute of Engineering Innovation, The University of Tokyo, 2−11−16 Yayoi, Bunkyo, Tokyo 113−8656, Japan. [3] Department of Materials Science and Engineering, Kyoto University, Yoshida-Honmachi, Sakyo, Kyoto 606−8501, Japan. [4] Department of Electrophysics, National Chiao Tung University, 1001, University Rd, Hsinchu 30010, Taiwan. [5] Center for Emergent Functional Matter Science, National Chiao Tung University, Hsinchu 30010, Taiwan. [6] Research Institute for Electronic Science, Hokkaido University, N20W10, Kita, Sapporo 001−0020, Japan. Correspondence and requests for materials should be addressed to H.O. (email: hiromichi.ohta@es.hokudai.ac.jp)

Currently, more than 60% of the energy produced from fossil fuels is lost as waste heat. Thermoelectric energy conversion, which is the process where waste heat is transformed into electricity by the Seebeck effect, is attracting attention as a potential energy harvesting technology[1–4]. The performance of thermoelectric materials is generally evaluated in terms of a dimensionless figure of merit,

$$ZT = S^2 \cdot \sigma \cdot T \cdot \kappa^{-1}, \qquad (1)$$

where $Z$ is the figure of merit, $T$ is the absolute temperature, $S$ is the thermopower (Seebeck coefficient), $\sigma$ is the electrical conductivity, and $\kappa$ is the sum of the electronic ($\kappa_{ele}$) and lattice thermal conductivities ($\kappa_{lat}$) of a thermoelectric material.

There are two strategies to improve $ZT$ of a thermoelectric material. One is to reduce $\kappa_{lat}$. Recently, state-of-the-art nanostructuring techniques have reduced $\kappa_{lat}$ significantly through phonon scattering by nanosized structural defects[5–8]. Such techniques have realized high-performance thermoelectric materials with a large $ZT$ of 1.5−2. The other strategy is an enhancement of the product $S^2 \cdot \sigma$, which is called power factor (PF). However, it is extremely difficult to enhance PF due to the trade-off relationship between $S$ and the carrier concentration ($n$). Therefore, PF has a local maximum value in three-dimensional (3D) bulk systems.

In a two-dimensional electron system (2DES) such as metal/insulator superlattices, electron carriers are confined within a thin layer (thickness thinner than the de Broglie wavelength, $\lambda_D$). 2DES is an efficient strategy to achieve an enhanced PF. The effectiveness of 2DES was theoretically predicted by Hicks and Dresselhaus[9]; 2DES in extremely narrow layers exhibits an enhanced $S$ without reducing $\sigma$ because the density of states (DOS) near the bottom of the conduction band increases as the 2DES layer thickness decreases. These layers are narrower than the $\lambda_D$,

$$\lambda_D = \frac{h}{\sqrt{3 \cdot m^* \cdot k_B \cdot T}}, \qquad (2)$$

where $h$, $m^*$, and $k_B$ are Planck's constant, effective mass of conductive electron or hole, and Boltzmann constant, respectively[9–13].

Many experimental studies have been made to clarify the effectiveness of 2DES to enhance PF using PbTe/Pb$_{1−x}$Eu$_x$Te multiple-quantum-well[10], electron-doped SrTiO$_3$-based superlattices[14,15],

SiGe-based superlattices[16,17], and Bi$_2$Te$_3$-based superlattices[18]. These 2DES layers showed enhanced $S$. However, total enhancement of PF was very small because of the insulator layer thickness. Thus, the effectiveness of 2DES has not been experimentally clarified thus far.

Here we experimentally clarify that an enhanced two-dimensionality is efficient to improve thermoelectric PF. We fabricated superlattices of [$N$ unit cell SrTi$_{1−x}$Nb$_x$O$_3$|11 unit cell SrTiO$_3$]$_{10}$—there are two different de Broglie wavelength in the SrTi$_{1−x}$Nb$_x$O$_3$ system. The maximum PF of the superlattice composed of the longer de Broglie wavelength SrTi$_{1−x}$Nb$_x$O$_3$ exceeded ~5 mW m$^{-1}$ K$^{-2}$, which doubles the value of optimized bulk SrTi$_{1−x}$Nb$_x$O$_3$. The present approach—use of longer de Broglie wavelength—is epoch-making and is fruitful to design good thermoelectric materials showing high PF.

## Results

**Hypothesis.** In order to enhance total PF of 2DES, two-dimensionality should be enhanced. Use of longer $\lambda_D$ should be effective if the electron carriers are confined within a defined thickness layer (Fig. 1). Very recently, we observed a steep decrease in $m^*/m_e$ at $x \sim 0.3$ in SrTiO$_3$–SrNbO$_3$ solid solution system, SrTi$_{1−x}$Nb$_x$O$_3$ ($x$ is ranging from 0.05 to 0.9; Fig. 2)[19]. The ratio $x$ of SrTi$_{1−x}$Nb$_x$O$_3$ can be divided into two regions, region A ($x$ is <0.3) and region B ($x$ is >0.3). The origin of the two regions is most likely due to the difference in the overlap population between the Ti 3$d$ and Nb 4$d$ orbitals ($r_{Ti3d}$ is 48.9 pm and $r_{Nb4d}$ is 74.7 pm)[20]. We calculated $\lambda_D$ values of SrTi$_{1−x}$Nb$_x$O$_3$ using the Eq. (2). The $\lambda_D$ value in region B is ~5.3 nm, which is 27% longer than that in region A (~4.1 nm). One can expect that $S$-enhancement factor in region B is much higher than that in region A because of higher two-dimensionality. Therefore, we hypothesized that SrTi$_{1−x}$Nb$_x$O$_3$-based 2DES can be used to clarify the effectiveness of 2DES to enhance PF experimentally.

We fabricated [$N$ uc SrTi$_{1−x}$Nb$_x$O$_3$|11 uc SrTiO$_3$]$_{10}$ super-lattices ($N$ is ranging from 1 to 12, $x$ is ranging from 0.2 to 0.9) by a pulsed laser deposition (PLD) technique on insulating (001) LaAlO$_3$ (pseudo-cubic perovskite, the lattice parameter, $a$ is 3.79 Å) single-crystal substrates using dense ceramic disks of a SrTiO$_3$–SrNbO$_3$ mixture and SrTiO$_3$ single crystal as the targets. The thicknesses of different layers were monitored in situ using the intensity oscillation of the reflection high-energy electron diffraction (RHEED) spots. (See Experimental Section.) High-

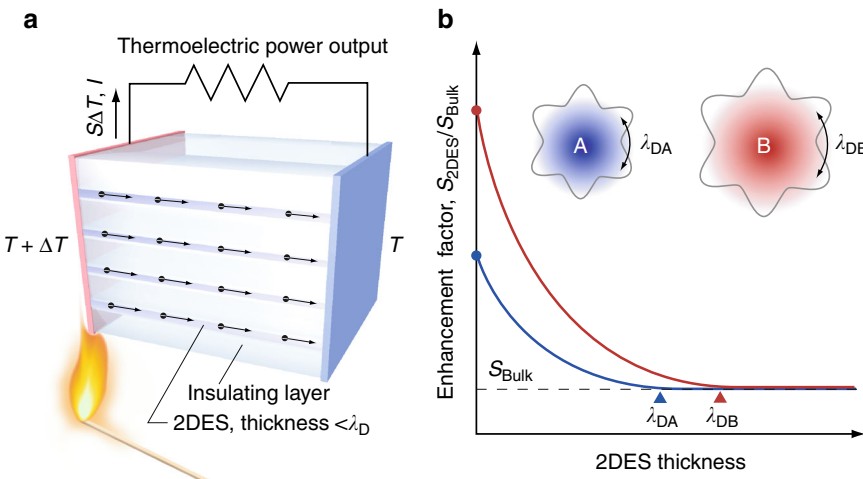

**Fig. 1** Thermoelectric effect of a 2D electron system. **a** Schematic illustration of thermoelectric Seebeck effect in a 2DES. A thermoelectric power output ($S \cdot \Delta T \cdot I$) can be obtained when $\Delta T$ is introduced. **b** The hypothesis that a 2DES with longer de Broglie wavelength ($\lambda_D$) shows a larger enhanced factor of thermopower

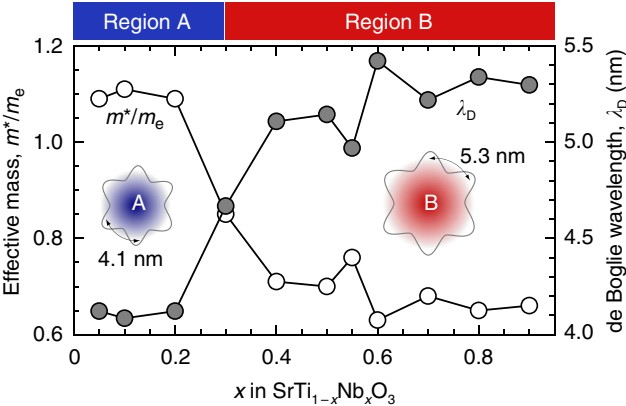

**Fig. 2** $SrTiO_3$–$SrNbO_3$ solid solution: a model system having two different $\lambda_D$. x-dependent effective mass ($m^*/m_e$, white symbols) and $\lambda_D$ (gray symbols) for $SrTi_{1-x}Nb_xO_3$ solid solutions. $m^*/m_e$ exerts a decreasing tendency with x, resulting in an increased $\lambda_D$. Sharp changes in both $m^*/m_e$ and $\lambda_D$ are detected around $x = 0.3$ due to the conduction band transition from Ti 3d to Nb 4d. The properties of $SrTi_{1-x}Nb_xO_3$ solid solutions can be divided into two regions based on the conduction bands (Ti 3d → region A and Nb 4d → region B). Inset: schematic illustrations of conduction electrons at regions A and B. At region B, $\lambda_D$ is ~5.3 nm, while it is ~4.1 nm at region A

resolution X-ray diffraction (XRD) measurements revealed that the resultant superlattices were heteroepitaxially grown on (001) $LaAlO_3$ with cube-on-cube epitaxial relationship with superlattice structure. Atomically smooth surfaces with stepped and terraced structure were observed by an atomic force microscopy (AFM).

**Microstructure and electronic structure**. Figure 3a summarizes the atomic arrangements of the [1 uc $SrTi_{0.4}Nb_{0.6}O_3$|11 uc $SrTiO_3$]$_{10}$ superlattice. Rather bright bands are observed near each $SrTi_{0.4}Nb_{0.6}O_3$ layer in the Cs-corrected high-angle annular dark-field scanning transmission electron microscopy (HAADF-STEM) image. In the magnified image, the #4 atom in the B-site column is brighter than the nearby atoms. However, there is no obvious difference in the A-site column, indicating Nb substitution occurs for the #4 atom in the B-site column. The electron energy loss spectroscopy signal of #4 is broader than that of the nearby atoms, implying the coexistence of $Ti^{4+}/Ti^{3+}$ in the $SrTi_{0.4}Nb_{0.6}O_3$ layers[21]. Therefore, in our superlattice fabrication, Nb ions are successfully confined into 1 uc of $SrTi_{0.4}Nb_{0.6}O_3$ layers[22].

In order to clarify the 2DES formation, the electronic band structures of the [1 uc $SrNbO_3$|10 uc $SrTiO_3$] superlattices were calculated based on the projector-augmented wave (PAW) method (Fig. 3b). The $E_F$ is located on the higher-energy side of the conduction band minimum for the first and second nearest-neighbor $SrTiO_3$ layers (Ti first NN and Ti second NN) together with the 1 uc $SrNbO_3$ layer (Nb). The electron carriers can seep from the $SrNbO_3$ layers into the $SrTiO_3$ layer. Delugas et al.[23] have also predicted theoretically that for lower Nb substituted samples, it is much easier for the electrons, especially in the $d_{xz}$ and $d_{yz}$ bands, to spread out to the neighboring $SrTiO_3$ layers, reducing the two-dimensionality. However, as the Nb content increases, the minimum thickness of the barrier layer may be reduced to 5 uc in the $SrNbO_3$ case. There is no doubt that the electron diffusion cannot be removed thoroughly in superlattice structure, but diffusion effects can be effectively suppressed by the high Nb substitution. From the band calculation, 2DES in our work is mainly confined to the 1 uc

$SrTi_{1-x}Nb_xO_3$ layers and should contribute to the S enhancement.

In order to further confirm the superlattice structure, we measured the $\kappa$ of the [1 uc $SrTi_{0.4}Nb_{0.6}O_3$|11 uc $SrTiO_3$]$_{10}$ superlattice along the cross-plane direction by time-domain thermal reflectance (TDTR) method. The total $\kappa$ could be suppressed to ~3.3 W m$^{-1}$ K$^{-1}$, similar to the minimum value of $CaTiO_3/SrTiO_3$-based superlattices ($\kappa$ ~ 3.2 W m$^{-1}$ K$^{-1}$) reported by Ravichandran et al.[24] From these results, we judged that our [N uc $SrTi_{1-x}Nb_xO_3$|11 uc $SrTiO_3$]$_{10}$ superlattices (N is ranging from 1 to 12, x is ranging from 0.2 to 0.9) are appropriate for us to clarify the effectiveness of 2DES to enhance PF.

**Thermoelectric properties**. The electrical conductivity ($\sigma$), carrier concentration ($n$), and Hall mobility ($\mu_{Hall}$) of the superlattices were measured at room temperature by a conventional d.c. four-probe method with a van der Pauw geometry. S was measured at room temperature by creating a temperature difference ($\Delta T$) of ~4 K across the film using two Peltier devices. Figure 4a summarizes the n-dependent S of [N uc $SrTi_{1-x}Nb_xO_3$|11 uc $SrTiO_3$]$_{10}$ superlattices (N is ranging from 1 to 12, x = 0.2, 0.3, and 0.8) along with bulk (~100-nm-thick $SrTi_{1-x}Nb_xO_3$ films, x = 0.2, 0.3, and 0.8, respectively) values for comparison. The bulk S for x = 0.2 was −143 μV K$^{-1}$, x = 0.3 was −73 μV K$^{-1}$, and x = 0.8 was −19 μV K$^{-1}$. The n value was measured based on the total thickness of the 2DES, which includes the insulating $SrTiO_3$ layers. All the 2DES samples show enhanced thermopower (−S) with a reduced N. Compared to the bulk samples at a similar n, a much higher −S is observed in superlattices as N is reduced below 3 uc.

To confirm the increasing two-dimensionality with x, the S-enhancement factors ($S_{2DES}/S_{Bulk}$) were plotted versus the N values (Fig. 4b). For 2DES with x = 0.2 and 0.3, the highest $S_{2DES}/S_{Bulk}$ values are around 4 and 5, respectively, whereas that for the x = 0.8 counterpart is ~10. As hypothesized, the enhanced $S_{2DES}/S_{Bulk}$ should stem from the increasing $\lambda_D$ with x. In our experiment, $S_{2DES}/S_{Bulk}$ for the x = 0.2 and 0.3 2DESs are saturated around 11 uc, which is consistent with $\lambda_D$ in region A (~4.2 nm indicated by dashed line $\lambda_{DA}$). As $\lambda_D$ increases in region B, the saturation position for the x = 0.8 2DES has a thickness larger than the $\lambda_D$ (~5.2 nm indicated by dashed line $\lambda_{DB}$). As a result, a significantly enhanced two-dimensionality is achieved in the x > 0.3 region B, which fits well with our hypothesis and suggests that region B has the potential to further enhance the thermoelectric PF.

Based on the conclusions above, we have enhanced thermoelectric PF in [1 uc $SrTi_{1-x}Nb_xO_3$|11 uc $SrTiO_3$]$_{10}$ superlattices by adjusting x between 0.2 and 0.9. Figure 5 summarizes the n dependences of the thermoelectric properties of [1 uc $SrTi_{1-x}Nb_xO_3$|11 uc $SrTiO_3$]$_{10}$ superlattices at room temperature along with the reported bulk values for comparison[19]. Following the bulk values, $\sigma$ increases almost linearly with n (Fig. 5a), indicating that n dominates $\sigma$. In the $SrTi_{1-x}Nb_xO_3$ system, carriers are mostly due to Nb substitution. The high n also induces a highly Nb substituted region with a superiority in $\sigma$. However, $\sigma$ for the superlattices remains lower than the bulk value due to the coexistence of 11 uc $SrTiO_3$ insulating layers.

$\mu_{Hall}$ for lower x of 2DESs (x ≤ 0.5) fluctuates around 3–5 cm$^2$ V$^{-1}$ s$^{-1}$, while for higher x 2DESs (x ≥ 0.6) values are ~6 cm$^2$ V$^{-1}$ s$^{-1}$ (Fig. 5b). Usually, $\mu_{Hall}$ is controlled by the conduction band of materials along with the effects of crystal defects such as impurities and grain boundaries. In the bulk samples, $\mu_{Hall}$ sharply increases due to the transition of the conduction band from Ti 3d to Nb 4d as x increases into the highly Nb substituted region[19]. This pattern is also observed in

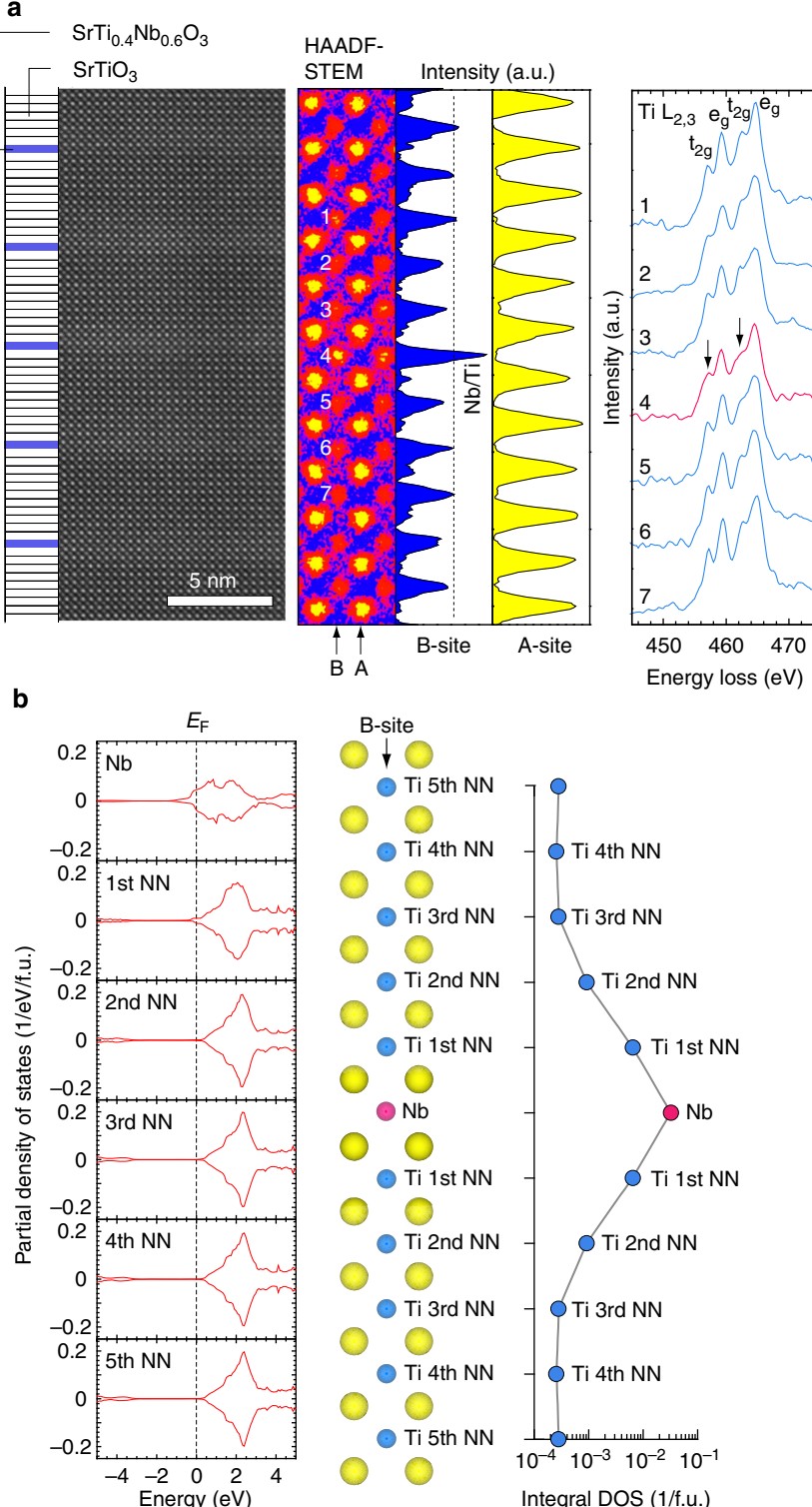

**Fig. 3** Experimental and theoretical analyses of the 2DES. **a** Cross-sectional HAADF-STEM image of the [1 uc $SrTi_{0.4}Nb_{0.6}O_3$|11 uc $SrTiO_3$]$_{10}$ superlattice. Layer stacking sequence is also shown. Rather bright bands are seen near each $SrTi_{0.4}Nb_{0.6}O_3$ layer. In the magnified image, the #4 atom in the B-site column is brighter than the nearby atoms, whereas no obvious difference is observed in the A-site column. EELS spectrum of #4 is broader than that of nearby atoms, indicating the coexistence of $Ti^{4+}/Ti^{3+}$ in the $SrTi_{0.4}Nb_{0.6}O_3$ layers. **b** The calculated partial DOS of Nb 4$d$ or Ti 3$d$ in the [1 uc $SrNbO_3$|10 uc $SrTiO_3$] superlattice. The Fermi energy ($E_F$) is located on the higher-energy side of the conduction band minimum for the first and second nearest neighbor (Ti first NN and Ti second NN). $SrTiO_3$ layers together with 1 uc $SrNbO_3$ layer (Nb) suggest that the electron carriers can seep from the $SrTi_{1-x}Nb_xO_3$ layers into the $SrTiO_3$ layers

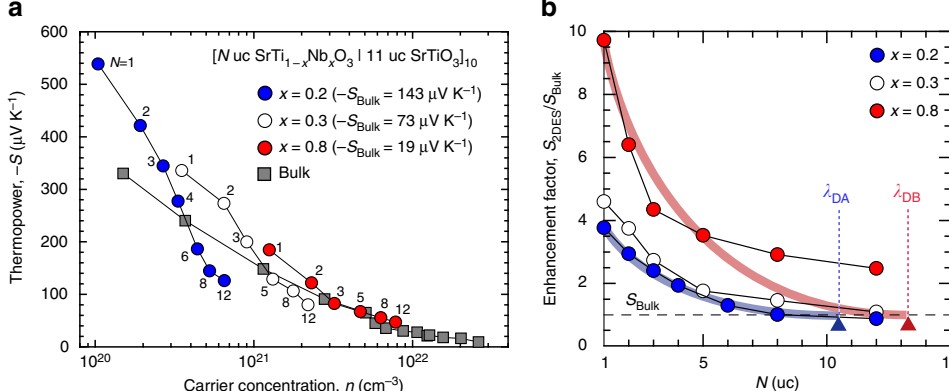

**Fig. 4** Two-dimensionality of 2DES: a key to enhance thermopower. **a** Plots of thermopower of the 2DESs, $[N$ uc $SrTi_{1−x}Nb_xO_3|11$ uc $SrTiO_3]_{10}$ superlattices ($x = 0.2$, 0.3, and 0.8), versus the carrier concentration ($n$). Compared to bulk values (gray squares), all the 2DESs show an enhanced $−S$ as $N$ is reduced under 3 uc. **b** Enhancement factors in $−S$ ($S_{2DES}/S_{Bulk}$) for three sets of 2DESs. For $x = 0.2$ and 0.3 2DESs, the highest $S_{2DES}/S_{Bulk}$ values are obtained at $N = 1$, which are 4 and 5, respectively, while that of $x = 0.8$ can reach 10

the superlattice counterparts. A higher $\mu_{Hall}$ ($\geq 6$ cm$^2$ V$^{-1}$ s$^{-1}$) is observed in samples with $x \geq 0.6$ than that for $x \leq 0.5$ ($3–5$ cm$^2$ V$^{-1}$ s$^{-1}$). Compared to the bulk samples, all the superlattices exert a much lower $\mu_{Hall}$, which may result from an insufficient crystal quality or electron diffusion into the pure SrTiO$_3$ barrier layers. Regardless, a conduction band transition from Ti $3d$ to Nb $4d$ is recognized in our superlattice systems. Due to the high overlapping population of the Nb $4d$ orbital, a superior electron transport property is realized in higher $x$ of 2DES.

Figure 5c plots the $S$ values for all the superlattices versus $n$ along with the reported bulk values[19]. The solid line depicts the overall tendency. In the diagram, the superlattices have a significantly enhanced $–S$ compared to bulk samples at similar $n$ values. As indicated by the solid lines, the experimental points for 2DES and bulk show different slopes of $–300$ and $–200$ μV K$^{-1}$ per decade, respectively. The relationship between $–S$ and $n_{eff}$ can be expressed by Eq. (3)

$$−S = −k_B/e \cdot \ln 10 \cdot A \cdot (\log n + B), \quad (3)$$

where $k_B$ is the Boltzmann constant and $e$ is an electron charge. $A$ and $B$ are the parameters that depend on the type of materials and their electronic band structures. Bulk shows a 3D electronic band structure with a parabolic shaped DOS near $E_F$, where the $A$ value = 1 and the slope reflects a constant value of $−k_B/e \cdot \ln 10$ ($−198$ μV K$^{-1}$). On the other hand, the slope of the 2DESs may reach $−300$ μV K$^{-1}$ per decade, indicating that the $A$ value = 1.5. Therefore, the 2DESs work well to enhance the $S$ even for the whole superlattice, including SrTiO$_3$ insulating layers.

Finally, we calculated PF of the $[1$ uc $SrTi_{1−x}Nb_xO_3|11$ uc $SrTiO_3]_{10}$ superlattices ($x$ is ranging from 0.2 to 0.9) using the observed $S$ and $\sigma$ values (Fig. 5d). PF is doubly enhanced for $x = 0.6$ ($5.1$ mW m$^{-1}$ K$^{-2}$ at $n \sim 8 \times 10^{20}$ cm$^{-3}$). Since the PF values are scattered due to the rather large distribution of $\mu_{Hall}$ ($3–6$ cm$^2$ V$^{-1}$ s$^{-1}$), we calculated PFs using the relationship between $S$ and $n$ (c) at constant $\mu_{Hall}$ ($6$ cm$^2$ V$^{-1}$ s$^{-1}$). The optimized PF of the 2DES should be $\sim 5$ mW m$^{-1}$ K$^{-2}$ at $n \sim 8 \times 10^{20}$ cm$^{-3}$, which doubles that of bulk SrTi$_{1−x}$Nb$_x$O$_3$ (PF $\sim 2.5$ mW m$^{-1}$ K$^{-2}$ at $n \sim 2 \times 10^{21}$ cm$^{-3}$).

## Discussion

The present 2DES, $[1$ uc $SrTi_{1−x}Nb_xO_3|11$ uc $SrTiO_3]_{10}$ superlattices ($x$ is ranging from 0.2 to 0.9), has several merits to enhance PF as compared with other 2DESs such as PbTe/Pb$_{1−x}$Eu$_x$Te multiple-quantum-well[10], SiGe-based superlattices[16,17], and Bi$_2$Te$_3$-based

superlattices[18], which are already commercialized thermoelectric materials. This is because SrTi$_{1−x}$Nb$_x$O$_3$ can be deposited with 1 uc layer accuracy by PLD. Therefore, we can easily reduce the 2DEG thickness to $\sim 0.4$ nm (1 uc layer). Further, there are two different $\lambda_D$ in SrTi$_{1−x}$Nb$_x$O$_3$; $\sim 4.1$ nm in the low conducting region and $\sim 5.3$ nm in the high conducting region. For enhancing PF, both $S$ and $\sigma$ play important roles. The present research implies that high conducting region is effective to enhance the thermoelectric PF in the 2DES. Herein highly Nb substitution are revealed to have the coexistence of both a high electron transport (high $n$ and $\mu_{Hall}$) and a high two-dimensionality (large $\lambda_D$).

In summary, we have experimentally clarified that an enhanced two-dimensionality of 2DES is efficient to improve thermoelectric PF. We measured the thermoelectric properties of 2DESs $[N$ uc $SrTi_{1−x}Nb_xO_3|11$ uc $SrTiO_3]_{10}$ superlattices ($N$ is ranging from 1 to 12, $x$ is ranging from 0.2 to 0.9) because there are two different $\lambda_D$ in this 2DES ($x > 0.3$: $\lambda_D \sim 5.3$ nm; $x < 0.3$: $\lambda_D \sim 4.1$ nm). The $S$-enhancement factor $S_{2DES}/S_{Bulk}$ of the 2DES ($N = 1$) for $x > 0.3$ were $\sim 10$, whereas those for $x < 0.3$ were $4–5$. Maximum PF of the 2DES ($N = 1$, $x = 0.6$) exceeded $\sim 5$ mW m$^{-1}$ K$^{-2}$, which doubles the value of optimized bulk SrTi$_{1−x}$Nb$_x$O$_3$ (PF $\sim 2.5$ mW m$^{-1}$ K$^{-2}$). The present 2DES approach—use of longer $\lambda_D$—is epoch-making and is fruitful to design good thermoelectric materials showing high PF.

## Methods

**Fabrication and analyses of the 2DESs**. A series of superlattices with the chemical formula of $[N$ uc $SrTi_{1−x}Nb_xO_3|11$ uc $SrTiO_3]_{10}$ ($N$ is ranging from 1 to 12, $x$ is ranging from 0.2 to 0.9) were fabricated by a PLD technique using dense ceramic disks of a SrTiO$_3$–SrNbO$_3$ mixture and a SrTiO$_3$ single crystal as the targets. The substrate was insulating (001) LaAlO$_3$ (pseudo-cubic perovskite, lattice parameter, $a$ is 3.79 Å, the surface area: 1 cm × 1 cm). The growth conditions were precisely controlled; the substrate temperature was 900 °C, the oxygen pressure was $\sim 10^{-4}$ Pa, and the laser fluence was $\sim 1.2$ J cm$^{-2}$ per pulse. The thicknesses of different layers were monitored in situ using the intensity oscillation of the RHEED spots. Details of our PLD growth process of the superlattices are reported elsewhere[14,25].

Crystallographic analyses of the resultant superlattices were performed by XRD (Cu Kα$_1$, ATX-G, Rigaku Co.), AFM (Nanocute, Hitachi Hi-Tech), and STEM (200 keV, JEM-ARM 200CF, JEOL Co. Ltd). TEM samples were fabricated using a cryo ion slicer (IB-09060CIS, JEOL Co. Ltd). HAADF images were taken with the detection angle of 68–280 mrad. Electron energy loss spectra were acquired in STEM mode with the energy resolution of 0.8 eV.

**Measurements of the thermoelectric properties of the 2DESs**. Electrical conductivity ($\sigma$), carrier concentration ($n$), and Hall mobility ($\mu_{Hall}$) were measured at room temperature by a conventional d.c. four-probe method with a van der Pauw geometry. $S$ was measured at room temperature by creating a temperature difference ($\Delta T$) of $\sim 4$ K across the film using two Peltier devices. (Two small

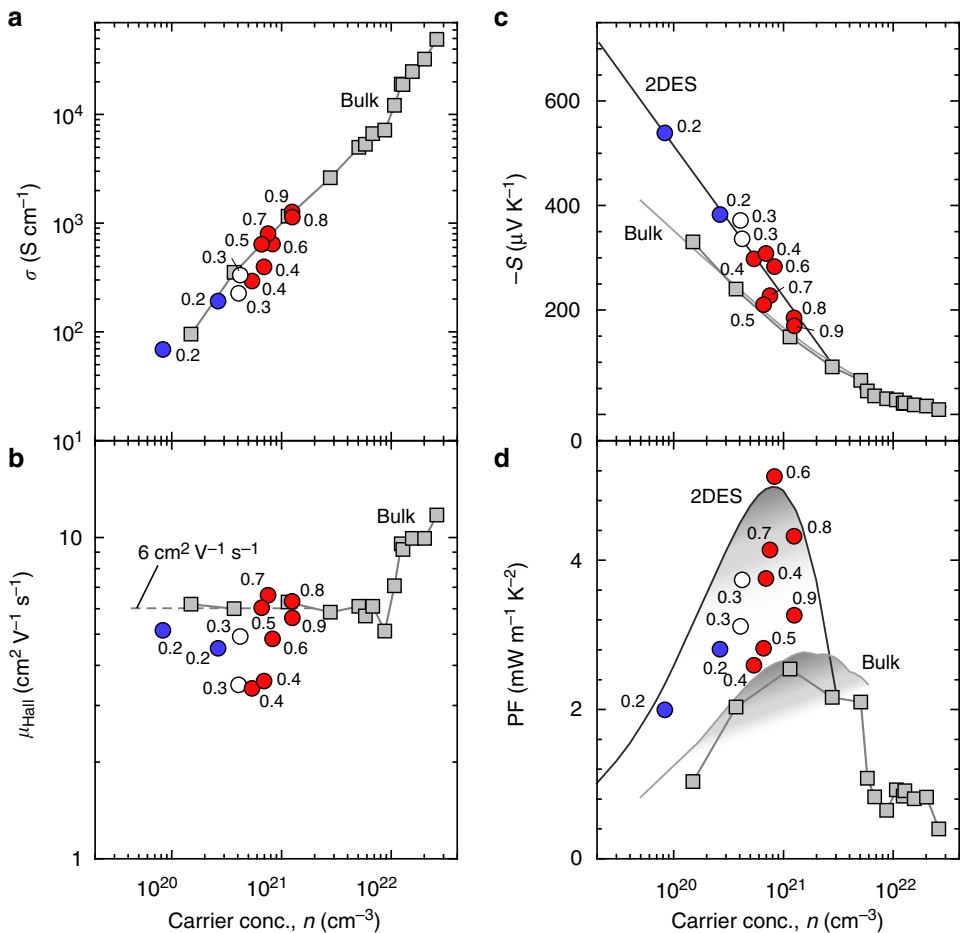

**Fig. 5** Double enhancement of the thermoelectric power factor in a 2DES. Carrier concentration dependences of **a** electrical conductivity ($\sigma$), **b** Hall mobility ($\mu_{Hall}$), **c** thermopower ($-S$), and **d** power factor [PF ($S^2 \cdot \sigma$)] of [1 uc SrTi$_{1-x}$Nb$_x$O$_3$ | 11 uc SrTiO$_3$]$_{10}$ 2DESs ($x$ is ranging from 0.2 to 0.9) at room temperature. Similar to the trends in the bulk values, $\sigma$ increases almost linearly with $n$. $\mu_{Hall}$ for lower $x$ samples ($x \leq 0.5$) fluctuates around 3–5 cm$^2$ V$^{-1}$ s$^{-1}$, while that for higher $x$ ones ($x \geq 0.6$) is ~6 cm$^2$ V$^{-1}$ s$^{-1}$. Slope of $-S$ versus log $n$ for bulk SrTi$_{1-x}$Nb$_x$O$_3$ is $-198$ $\mu$V K$^{-1}$, which is ~1.5 times lower than $-300$ $\mu$V K$^{-1}$ for the 2DESs. Double enhancement of PF is seen in $x = 0.6$ (5.1 mW m$^{-1}$ K$^{-2}$ at $n \sim 8 \times 10^{20}$ cm$^{-3}$). Since the PF values are scattered due to the rather large distribution of $\mu_{Hall}$ (3–6 cm$^2$ V$^{-1}$ s$^{-1}$), we calculated PFs using the relationship between $S$ and $n$ (c) at constant $\mu_{Hall}$ (6 cm$^2$ V$^{-1}$ s$^{-1}$). The optimized PF of the 2DES should be ~5 mW m$^{-1}$ K$^{-2}$ at $n \sim 8 \times 10^{20}$ cm$^{-3}$, which doubles that of bulk SrTi$_{1-x}$Nb$_x$O$_3$ (PF $\sim$ 2.5 mW m$^{-1}$ K$^{-2}$ at $n \sim 2 \times 10^{21}$ cm$^{-3}$)

thermocouples were used to monitor the actual temperatures of each end of a superlattice.) The thermo-electromotive force ($\Delta V$) and $\Delta T$ were measured simultaneously, and the $S$ values were obtained from the slope of the $\Delta V$–$\Delta T$ plots (the correlation coefficient: >0.9999).

Cross-plane thermal conductivity ($\kappa$) was measured by TDTR (Picotherm Co.) method. Mode-locked fiber pulse lasers with 1550 and 775 nm wavelengths were used for heating and measuring, respectively. Both lasers are with the repetition frequency of 20 MHz and pulse duration of 0.4 ps. Before measurement, Mo film with a thickness of 100 nm was first deposited on the surface of the sample as the transducer. During measurement, time-dependent transient thermoreflectance phase signal of Mo transducer was measured, from which $\kappa$ was further simulated. Time-domain thermoreflectance was measured based on amplified laser systems (5 kHz and ~200 fs centered at 1030 nm). Degenerate pump and probe photons were separated by the cross polarization, and a polarizing filter was employed before the lock-in detection. A mechanical delay stage was used for time scan up to 1.5 ns. Pump to probe intensity ratio was >15, and the size ratio was around 6.

**Energy band calculation of the 2DES.** Band structure for the [1 uc SrNbO$_3$|10 uc SrTiO$_3$] superlattice was calculated based on the PAW method[26], as implemented in the VASP code[27,28]. We adopted the Heyd–Scuseria–Ernzerhof hybrid functionals[29–31] and a plane-wave cutoff energy of 550 eV. 6 × 6 × 6 and 6 × 6 × 2 k-point meshes were employed in the total-energy evaluations and geometry optimization for the perovskite unit cells of SrTiO$_3$ and the superlattice cell, respectively. The in-plane lattice constant of the superlattice cell was fixed at the optimized value of SrTiO$_3$ while the out-of-plane lattice constant and the atomic coordinates were fully relaxed.

**Data availability**. The data that support the findings of this study are available from the corresponding authors upon reasonable request.

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

## Acknowledgements

This research was supported by Grants-in-Aid for Scientific Research on Innovative Areas "Nano Informatics" (25106003, 25106005, and 25106007) from the Japan Society for the Promotion of Science (JSPS). H.O. was supported by Grants-in-Aid for Scientific Research A (17H01314) and B (26287064) from the JSPS. Y.Z. thanks to the China Scholarship Council (CSC) for a scholarship to study in Japan. Y.M.S. acknowledge the grant from Taiwan Ministry of Science and Technology (MOST 104-2112-M-009-023-MY3 and MOST 104-2738-M-009-006), and the Center for Emergent Functional Matter Science of National Chiao Tung University from The Featured Areas Research Center Program within the framework of the Higher Education Sprout Project by the Ministry of Education (MOE) in Taiwan. A part of this work was supported in part by the Network Joint Research Center for Materials and Devices and by Dynamic Alliance for Open Innovation Bridging Human, Environment and Materials.

## Author contributions

Y.Z. performed the sample preparation and measurements. B.F. and Y.I. performed the STEM analyses. H.H. and I.T. performed the energy band calculations. C.P.C. and Y.M.S. provided measurements on the time-domain thermoreflectance. H.O. planned and supervised the project. All authors discussed the results and commented on the manuscript.

## Additional information

**Competing interests:** The authors declare no competing interests.

