## [Peer Review File · Nature Communications]

Reviewers' comments:

Reviewer #1 (Remarks to the Author):

This manuscript presented a very interesting experimental result regarding to the quantum confinement effect in a superlattice system of $[\text{Nuc SrTi}_{1-x}\text{Nb}_x\text{O}_3 - 11\text{uc SrTiO}_3]_{10}$. The results come out that both composition and the N value play an important role on the thermopower of superlattice samples, which was majorly due to the adjustment of the de Broglie wavelength and carrier effective mass by the Nb content and the variation of film thickness. The sample of $[1\text{uc SrTi}_{0.4}\text{Nb}_{0.6}\text{O}_3 - 11\text{uc SrTiO}_3]_{10}$ possess the highest power factor of about $5 \text{ mWm}^{-1}\text{K}^{-2}$, which doubled the value of the bulk counterpart. The referee recommends the publication of this manuscript in Nature communication unless the clear and appropriate response to the referee's comments listed below.

It is obvious the thermopower is decreasing with the increasing N (shown in Fig. 4(a)), which is due to the decreasing ratio of $\lambda_{\text{DB}}/\text{thickness}$. This should imply the quantum confinement effect got involved in the evolution of the thermopower. However, the referee does not quite understand the following three points.

1. The authors compared the $S_{2\text{DES}}$ with S_{bulk} . Did the authors compare the same composition? If not, what will be the selection scenario or what is the composition list for the bulk counterpart?
2. For the sample of $[1\text{uc SrTi}_{0.4}\text{Nb}_{0.6}\text{O}_3 - 11\text{uc SrTiO}_3]_{10}$, we can see from Fig. 5 that the conductivity is lower than the bulk (see Fig. 5(a)) and the Seebeck is roughly 50% higher than the bulk, resulting in a doubled power factor. How is the measurement error? Could the authors provide more detail on the calibration of the measuring equipment?
3. The author should also clearly identify the relationship among the thickness, the carrier concentration, the carrier mobility and the Seebeck coefficient. E. I. Rogacheva et al. described an oscillation of carrier concentration and Seebeck coefficient with film thickness (APPLIED PHYSICS LETTERS 106, 053103 (2015)). More detailed discussion and some simulation in addition to the theoretical calculation on band structure should also be emphasized in this study.

Reviewer #2 (Remarks to the Author):

The thermoelectric transport behavior in two electron systems is very attractive in both fields of physics and materials. I read the manuscript with great interests. Three concerns/comments are raised about this manuscript.

(1) Figure 1 is too schematic and may be not suitable as a result figure. Figure 1(a) does not provide new scientific information. Figure 1(b) seems to be a schematic based on hypothesis. This paper is a communication focusing on experimental results but not a review to introduce concept.

(2) Recently, I found some papers (e.g. ACS Energy Letters, by Jikun Chen) on the thermoelectric properties of $\text{SrNb}_{0.2}\text{Ti}_{0.8}\text{O}_3$ oxide thin film on a single crystal substrate. In those newly published papers, it was reported that the lattice strain is likely attributed to the enhancement of thermoelectric power factors. At least, by my view, there is some similarity between present manuscript and the recently published work. However, the authors did not mention those results and did not discuss the possibility of influence of lattice strain. I suggest the authors to analyze/discuss the lattice strain in the superlattice films.

(3) I also suggest the authors to show the details of the TEM observation and the composition distribution analysis and then check the interdiffusion across the interface in the superlattice films.

Reviewer #3 (Remarks to the Author):

The manuscript reports the enhancement of thermoelectric power factor of a 2D electron system by adjusting the thickness of 2DES and de Broglie wavelength. Combining experimental and theoretical analyses on 2DES, the authors concluded that the thermoelectric power factor can be largely improved by increasing two-dimensionality of 2DES. The experimental and theoretical results are of good quality, however, the analysis and study as presented has a number of problems which I believe must be clarified and improved before the work can be considered for publication.

Comments not in order of significance:

1. In Figure 4b, to confirm the increasing two-dimensionality with x , the S-enhancement factors (S_{2DES}/S_{Bulk}) were plotted versus the N values. It is necessary for authors to clarify which sample is

used to obtain the S_{Bulk} . Obviously, the $\text{SrTi}_{1-x}\text{Nb}_x\text{O}_3$ or SrTiO_3 bulk sample were unsuitable used to determine the the S -enhancement factors. In my opinion, the pure SrTiO_3 and $\text{SrTi}_{1-x}\text{Nb}_x\text{O}_3$ film without superlattices should be used to determine the the S -enhancement factors of 2DES.

2.Regarding to the theoretical analyses of the 2DES, the $[1 \text{ uc SrNbO}_3 | 10 \text{ uc SrTiO}_3]$ superlattice was adopted to calculate partial DOS of Nb 4d or Ti 3d. However, the $[1 \text{ uc SrNbO}_3 | 10 \text{ uc SrTiO}_3]$ superlattice is different from the actual $[N \text{ uc SrTi}_{1-x}\text{Nb}_x\text{O}_3 | 11 \text{ uc SrTiO}_3]_{10}$ superlattice. Will the theoretical calculation based on the structural model as shown in Figure 3 (b) reflect the electronic band structure of virtual structure?

3.To increase the power factor is certainly a effective way to improve figure of merit of a thermoelectric material, however, the figure of merit is also depends on the thermal conductivity of thermoelectrics. Given that the enhancement of thermoelectric power factor of 2DES is accompanied with the increase in electrical conductivity as indicated in Figure 5(a), the electronic thermal conductivity will be changed accordingly. Therefore, it is necessary for authors to discuss the change of thermal conductivity so that the performance of thermoelectric material can be systematically evaluated.

4.By varying the composition of SrTiO_3 - SrNbO_3 solid solution, the authors obtained two regions with two different de Broglie wavelengths. Within region B, the de Broglie wavelength almost does not change much with x in $\text{SrTi}_{1-x}\text{Nb}_x\text{O}_3$ as shown in Figure 2, however the large difference in PF is observed for $x=0.6$ and $x=0.9$ as shown in Figure 5 (d). If the enhancement of power factor is attributed to the variation of de Broglie wavelengths, how to explain the observed difference in PF with almost similar de Broglie wavelengths in region B.

5.In page 5, it is described that "EELS spectrum of #4 is broader than that of nearby atoms, indicating the co-existence of $\text{Ti}^{4+}/\text{Ti}^{3+}$ ". Whereas the the EELS spectrum of #2 is also broad compared to EELS spectrum of other atoms, does it mean that the coexistence of $\text{Ti}^{4+}/\text{Ti}^{3+}$ is also take place in neighboring layers of atom 4.

Reviewer #1

This manuscript presented a very interesting experimental result regarding to the quantum confinement effect in a superlattice system of [N uc $\text{SrTi}_{1-x}\text{Nb}_x\text{O}_3$ – 11 uc SrTiO_3] $_{10}$. The results come out that both composition and the N value play an important role on the thermopower of superlattice samples, which was majorly due to the adjustment of the de Broglie wavelength and carrier effective mass by the Nb content and the variation of film thickness. The sample of [1 uc $\text{SrTi}_{0.4}\text{Nb}_{0.6}\text{O}_3$ – 11 uc SrTiO_3] $_{10}$ possess the highest power factor of about $5 \text{ mW m}^{-1} \text{ K}^{-2}$, which doubled the value of the bulk counterpart. The referee recommends the publication of this manuscript in *Nature Communications* unless the clear and appropriate response to the referee's comments listed below.

It is obvious the thermopower is decreasing with the increasing N (shown in Fig. 4(a)), which is due to the decreasing ratio of λ_{DA} /thickness. This should imply the quantum confinement effect got involved in the evolution of the thermopower. However, the referee does not quite understand the following three points.

Thank you very much for taking your time to review our manuscript. We are really happy to hear your positive opinion. It is such an honor to be able to have you say such a thing.

Comment 1

The authors compared the S_{2DES} with S_{bulk} . Did the authors compare the same composition? If not, what will be the selection scenario or what is the composition list for the bulk counterpart?

Yes, you are right. We compared the same composition. In order to prevent the misreading, we added following as the explanation of the bulk.

(~100-nm-thick $\text{SrTi}_{1-x}\text{Nb}_x\text{O}_3$ films with $x=0.2, 0.3,$ and $0.8,$ respectively)

Comment 2

For the sample of [1 uc $\text{SrTi}_{0.4}\text{Nb}_{0.6}\text{O}_3$ – 11 uc SrTiO_3] $_{10}$, we can see from Fig. 5 that the conductivity is lower than the bulk (see Fig. 5(a)) and the Seebeck is roughly 50% higher than the bulk, resulting in a doubled power factor. How is the measurement error? Could the authors provide more detail on the calibration of the measuring equipment?

Research Institute for Electronic Science, Hokkaido University, N20W10, Kita, Sapporo 001-0020, JAPAN

Our films are enough large ($1\text{ cm} \times 1\text{ cm}$) to measure the thermopower. We could add temperature difference more than $\pm 4\text{ K}$ and obtained quite a nice linearity of 0.9999, which means that measurement error is negligibly small. At least, we do not need to use error bar. We added followings in the Methods section.

The substrate was insulating (001) LaAlO_3 (pseudo-cubic perovskite, $a=3.79\text{ \AA}$, **The surface area: $1\text{ cm} \times 1\text{ cm}$**).

The thermo-electromotive force (ΔV) and ΔT were measured simultaneously, and the S -values were obtained from the slope of the ΔV - ΔT plots **(The correlation coefficient: >0.9999)**.

Comment 3

The author should also clearly identify the relationship among the thickness, the carrier concentration, the carrier mobility and the Seebeck coefficient. E. I. Rogacheva *et al.* described an oscillation of carrier concentration and Seebeck coefficient with film thickness [*Appl. Phys. Lett.* 106, 053103 (2015)]. More detailed discussion and some simulation in addition to the theoretical calculation on band structure should also be emphasized in this study.

Thank you for the information. We did not know that Bi_2Te_3 thin film shows very interesting oscillation behaviour of thermopower. However, our superlattice films do not show any thickness dependence of carrier concentration, mobility and thermopower.

Research Institute for Electronic Science, Hokkaido University, N20W10, Kita, Sapporo 001-0020, JAPAN

Reviewer #2

The thermoelectric transport behavior in two electron systems is very attractive in both fields of physics and materials. I read the manuscript with great interests. Three concerns/comments are raised about this manuscript.

Thank you very much for taking your time to review our manuscript. We are really happy to hear your positive opinion.

Comment 1

Figure 1 is too schematic and may be not suitable as a result figure. Figure 1(a) does not provide new scientific information. Figure 1(b) seems to be a schematic based on hypothesis. This paper is a communication focusing on experimental results but not a review to introduce concept.

In order to explain what we did to the readers of *Nature Communications*, who are mostly not specialist, we are thinking the Figs. 1a and 1b are necessary for this manuscript. Since this manuscript is not a Communication but an Article, we should introduce our concept of this research.

Comment 2

Recently, I found some papers (e.g. *ACS Energy Letters*, by Jikun Chen) on the thermoelectric properties of $\text{SrNb}_{0.2}\text{Ti}_{0.8}\text{O}_3$ oxide thin film on a single crystal substrate. In those newly published papers, it was reported that the lattice strain is likely attributed to the enhancement of thermoelectric power factors. At least, by my view, there is some similarity between present manuscript and the recently published work. However, the authors did not mention those results and did not discuss the possibility of influence of lattice strain. I suggest the authors to analyze/discuss the lattice strain in the superlattice films.

Since the superlattice films are fully relaxed on (001) LaAlO_3 substrate, we do not need to discuss the possibility of influence of lattice strain.

Comment 3

Research Institute for Electronic Science, Hokkaido University, N20W10, Kita, Sapporo 001-0020, JAPAN

I also suggest the authors to show the details of the TEM observation and the composition distribution analysis and then check the interdiffusion across the interface in the superlattice films.

Thank you very much for your helpful comment. We have already confirmed that Nb can be confined to a one-unit-cell thickness without interdiffusion in SrTiO₃. Such result is also in good agreement with our previous experimental and theoretical studies. [T. Mizoguchi, H. Ohta, H. Lee, N. Takahashi, and Y. Ikuhara, *Adv. Funct. Mater.* **21**, 2258 (2011).] We added this paper as reference 22.

Reviewer #3

The manuscript reports the enhancement of thermoelectric power factor of a 2D electron system by adjusting the thickness of 2DES and de Broglie wavelength. Combining experimental and theoretical analyses on 2DES, the authors concluded that the thermoelectric power factor can be largely improved by increasing two-dimensionality of 2DES. The experimental and theoretical results are of good quality, however, the analysis and study as presented has a number of problems which I believe must be clarified and improved before the work can be considered for publication.

Thank you very much for taking your time to review our manuscript. We are really happy to hear your positive opinion.

Comment 1

In Figure 4b, to confirm the increasing two-dimensionality with x , the S -enhancement factors (S_{2DES}/S_{Bulk}) were plotted versus the N values. It is necessary for authors to clarify which sample is used to obtain the S_{Bulk} . Obviously, the $SrTi_{1-x}Nb_xO_3$ or $SrTiO_3$ bulk sample was unsuitable used to determine the S -enhancement factors. In my opinion, the pure $SrTiO_3$ and $SrTi_{1-x}Nb_xO_3$ film without superlattices should be used to determine the S -enhancement factors of 2DES.

We used ~ 100 -nm-thick $SrTi_{1-x}Nb_xO_3$ films grown on (001) $LaAlO_3$ substrate to determine the S -enhancement factors of 2DES. In order to prevent the misreading, we added following as the explanation of the bulk.

(~ 100 -nm-thick $SrTi_{1-x}Nb_xO_3$ films with $x=0.2, 0.3,$ and $0.8,$ respectively)

Comment 2

Regarding to the theoretical analyses of the 2DES, the [1 uc $SrNbO_3$ |10 uc $SrTiO_3$] superlattice was adopted to calculate partial DOS of Nb 4d or Ti 3d. However, the [1 uc $SrNbO_3$ |10 uc $SrTiO_3$] superlattice is different from the actual [N uc $SrTi_{1-x}Nb_xO_3$ |11 uc $SrTiO_3$]₁₀ superlattice. Will the theoretical calculation based on the structural model as shown in Figure 3 (b) reflect the electronic band structure of virtual structure?

The difference between the virtual structure as shown in **Fig. 3a** (the [1 uc $\text{SrTi}_{0.4}\text{Nb}_{0.6}\text{O}_3$ | 10 uc SrTiO_3] superlattice) and the calculation model (the [1 uc SrNbO_3 | 10 uc SrTiO_3] superlattice) is a few Nb concentration in the SrNbO_3 monolayer. This is insignificant difference since our discussion focuses the range of the electron exudation from the SrNbO_3 layer to SrTiO_3 layers.

Comment 3

To increase the power factor is certainly an effective way to improve figure of merit of a thermoelectric material, however, the figure of merit is also depends on the thermal conductivity of thermoelectrics. Given that the enhancement of thermoelectric power factor of 2DES is accompanied with the increase in electrical conductivity as indicated in Figure 5(a), the electronic thermal conductivity will be changed accordingly. Therefore, it is necessary for authors to discuss the change of thermal conductivity so that the performance of thermoelectric material can be systematically evaluated.

We measured the PF in the in-plane direction of the superlattices. Therefore, we need the thermal conductivity in the in-plane direction to discuss about ZT , but it is almost impossible to measure. That is the reason why we have not discussed the ZT .

Comment 4

By varying the composition of SrTiO_3 – SrNbO_3 solid solution, the authors obtained two regions with two different de Broglie wavelengths. Within region B, the de Broglie wavelength almost does not change much with x in $\text{SrTi}_{1-x}\text{Nb}_x\text{O}_3$ as shown in Figure 2, however the large difference in PF is observed for $x=0.6$ and $x=0.9$ as shown in Figure 5 (d). If the enhancement of power factor is attributed to the variation of de Broglie wavelengths, how to explain the observed difference in PF with almost similar de Broglie wavelengths in region B.

The PF is given by $S^2 \cdot n \cdot \mu \cdot e$. Since $|S|$ decreases with increasing n , the PF shows a peak as show in Fig. 5d (solid line).

Comment 5

In page 5, it is described that “EELS spectrum of #4 is broader than that of nearby atoms, indicating the co-existence of $\text{Ti}^{4+}/\text{Ti}^{3+}$ ”. Whereas the EELS spectrum of #2

is also broad compared to EELS spectrum of other atoms, does it mean that the coexistence of $\text{Ti}^{4+}/\text{Ti}^{3+}$ is also take place in neighboring layers of atom 4.

Thank you very much for your careful reading. As you pointed out, the peaks of #2 look broader, however, the difference of #2 and #4 can be still clearly seen by using the following spectra.

If you compare the spectra of #2 and #4, there is a clear difference in the peak intensity of Ti L_2 - t_{2g} as indicated by arrows. The splitting features of t_{2g} and e_g in Ti L_2 edge are clearly visible in #2 compared to spectrum #4. On the other hand, if you compare the spectra of #2 and #1, the peak intensity of Ti L_2 - t_{2g} is almost the same. The difference clearly supports the description that #4 is broader than that of nearby atoms, indicating the co-existence of $\text{Ti}^{4+}/\text{Ti}^{3+}$.

REVIEWERS' COMMENTS:

Reviewer #2 (Remarks to the Author):

The revised version reached acceptable level.

Reviewer #3 (Remarks to the Author):

The authors have clarified all questions and revised the manuscript in accordance with the comments from referees. The revised version of manuscript can be recommended for publication in Nature Communications.